# Low-Computation Adaptive Saturated Self-Triggered Tracking Control of Uncertain Networked Systems

**Wenjing Wu** [1], **Ning Xu** [2,*], **Ben Niu** [3], **Xudong Zhao** [4] and **Adil M. Ahmad** [5]

1 College of Control Science and Engineering, Bohai University, Jinzhou 121013, China; wenjingwuyh@163.com
2 College of Information Science and Technology, Bohai University, Jinzhou 121013, China
3 School of Information Science and Engineering, Shandong Normal University, Jinan 250014, China; niubensdnu@163.com
4 Faculty of Electronic Information and Electrical Engineering, Dalian University of Technology, Dalian 116024, China; xdzhaohit@gmail.com
5 Communication Systems and Networks Research Group, Department of Information Technology, Faculty of Computing and Information Technology, King Abdulaziz University, Jeddah 22254, Saudi Arabia; aahmad@kau.edu.sa
* Correspondence: hpxuning@163.com

**Abstract:** In this paper, a low-computation adaptive self-triggered tracking control scheme is proposed for a class of strict-feedback nonlinear systems with input saturation. By introducing two novel error transformation functions, the designed low-computation adaptive control scheme can overcome the problem of complexity explosion in the absence of any filters, such that the developed control scheme is more applicable. In addition, to save communication resources in networked systems, a self-triggered communication strategy is proposed which can predict the next trigger point based on the current information. Compared with traditional event-triggered mechanisms, the computational burden arising from continuous monitoring of threshold conditions was successfully avoided. Furthermore, the input saturation problem considered in this paper prevents the overload phenomenon caused by signal jumps, and the adverse effects are compensated by introducing an auxiliary system. The effectiveness of the developed control scheme is verified through a simulation example.

**Keywords:** low-computation technology; self-triggered control; tracking control; input saturation; prescribed performance

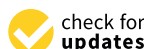



## 1. Introduction

In the past few decades, the control of nonlinear systems has become an increasingly popular research topic across various fields [1–3]. To date, several control methods have been proposed to address challenges posed by nonlinear dynamics, including robust control [4], sliding mode control [5,6], adaptive control [7,8], etc. Among these methods, adaptive backstepping control has been widely acknowledged as one of the effective approaches to handling nonlinear dynamics in systems. In addition, the presence of uncertain nonlinear dynamics can also affect system performances. In this case, addressing the impact of uncertain dynamics on systems becomes a prerequisite for adaptive backstepping control design. To overcome this challenge, fuzzy logic systems (FLSs) [9–11] and neural networks (NNs) [12–14] have been introduced in control design to deal with uncertain nonlinearity, primarily due to their outstanding approximation ability. In this way, adaptive backstepping control for uncertain nonlinear systems has achieved rapid developments. In [15], an adaptive control method exploring radial basis function neural networks (RBF NNs) was developed for a class of uncertain nonlinear systems under additional disturbances, where the uncertain dynamics in systems are linearized through the approximation ability of RBF NNs. In [16], based on the backstepping design framework and improved FLSs, an adaptive fuzzy compensation controller is established to handle actuator failures and dead-zone

constraints that occur in uncertain nonlinear systems. Although backstepping technology is an important tool for addressing control design problems of nonlinear systems, under the traditional backstepping design framework, the large computational burden has become an important drawback, limiting its wide application.

Specifically, in traditional backstepping technology, the derivation of virtual control signals becomes increasingly burdensome as the number of system orders increases, eventually leading to the problem of complexity explosion, and this will result in extremely high complexity for the final controller. Aiming at this problem, the command-filtered strategy was developed by introducing a first-order low-pass filter and designing corresponding filtering compensation signals to reduce filtering errors [17–20]. The authors in [21] proposed an adaptive output feedback control strategy by using command filtering and backstepping technology to address the problem of complexity explosion. Unfortunately, with the advent of specific filters, although the problem of complexity explosion was successfully solved, the structure of controllers also became more complex, and filtered compensation signals imposed some additional computational burdens. Based on this situation, in this paper, by introducing a low-computation technology, the computational burdens generated by the backstepping method, command filters, a large number of adaptive parameters, etc, are overcome. Currently, low-computation technology has been applied widely in nonlinear systems. The authors in [22] proposed a low-computation adaptive control method based on prescribed performance, which greatly reduced the computational burden of a system. The tracking control problem for strict-feedback nonlinear systems with unmatched disturbances was considered in [23], where, combined with constraint-handling techniques, a low-computation adaptive fuzzy control strategy was developed. Furthermore, in conventional control schemes, the control signals are updated according to a specific period sampling time, which leads to a large amount of data occupying the communication channel and increases the communication pressure.

Efficient utilization of communication resources is crucial for optimizing the performances of control systems. In the current networked control context, signal transmissions between a controller and an actuator are achieved by sampling a shared communication channel [24]. Despite this, a large number of signals are generated using time-sampling methods, yet the available communication channel bandwidth is usually limited, which exacerbates the communication pressure. To solve the above problems, event-triggered [25–27] and self-triggered control strategies [28–30] are presented to reduce the amount of information transmissions in communication processes. In [31], an event-triggered mechanism was incorporated into the design of an adaptive control scheme for a category of uncertain nonlinear systems, with the aim of conserving communication resources, where the next trigger point was established by devising a suitable trigger condition (threshold). Therein, the information can be passed to the controller only when the condition is satisfied; otherwise, the current information is discarded. In addition, the event-triggered mechanism requires continuously monitoring signals, which is difficult to achieve in actual systems. Based on this situation, we introduced a self-triggered mechanism to improve it. Differently from traditional event-triggered strategies, the self-triggered control scheme predicts the next trigger point through the current system sampling information, thereby avoiding the need for continuously monitoring system signals.

On the other hand, in practical engineering, input saturation often occurs in amplification and actuator components, which can degrade system performances and even lead to system instability. As a result, the input saturation problem for nonlinear systems is challenging, and it has received a lot of attention. In [32], the authors proposed an adaptive fuzzy control scheme for a class of uncertain non-strict-feedback nonlinear systems with input saturation, where the input saturation problem was solved by introducing an auxiliary design system. In [33], a multigradient recursive reinforcement learning scheme for discrete-time nonlinear systems with input saturation was proposed. In [34], the author presented an observer-based adaptive fuzzy output feedback control strategy for a category of uncertain nonlinear systems with input saturation and output constraints which were

prone to unforeseen states, and the designed controller effectively addressed the impact of input saturation and output constraints. Therefore, in the case of reducing the calculation complexity, designing an adaptive self-triggering control scheme that considers both communication resources and input saturation has become a difficult problem.

Motivated by the above discussion, this paper develops a low-computation adaptive self-triggered control strategy for a class of uncertain nonlinear systems with input saturation. The designed scheme avoids the problem of complexity explosion and improves the transmission efficiency of networked systems. The contributions of this paper in comparison with the existing literature are listed below:

1.  Compared with the existing literature [17–21], the adaptive low-computation control strategy designed in this paper avoids the problem of complexity explosion and reduces the computational burden of a system without introducing any filters.
2.  To save communication resources, a self-triggered mechanism is designed in this paper which can predict the next trigger point based on the current system information, avoiding the problem of continuous monitoring of thresholds in an event-triggered mechanism [25–27] and greatly improving the transmission efficiency of a system.
3.  When the input signal approaches the saturation limit, an auxiliary system is introduced to produce a compensation signal, which reduces the saturation effects and maintains system performances.

## 2. Problem Formulation and Preliminaries

### 2.1. System Description

The majority of engineering systems, such as compressors for jet engines, biochemical processes, active suspension systems, single-link flexible robots, etc., can be converted into strict-feedback forms. The following strict-feedback nonlinear systems are taken into consideration:

$$\dot{x}_i = \Phi_i(\bar{x}_i) + x_{i+1}, i = 1, \ldots, n-1$$
$$\dot{x}_n = \Phi_n(\bar{x}_n) + u(\varpi) \tag{1}$$
$$y = x_1$$

where $y \in \mathbb{R}$ and $\bar{x}_i = [x_1, \ldots, x_i]^T \in \mathbb{R}^i$ represent the measured the system output and state vectors, respectively. $\Phi_i(\bar{x}_i)$ is the unknown nonlinear functions. $\mathbb{R}^i \to \mathbb{R}$ is locally Lipschitz in $\bar{x}_i$ [35]. $u(\varpi)$ is the saturation input to system (1) and is represented as such

$$u[\varpi(t)] = sat[\varpi(t)] = \begin{cases} sign[\varpi(t)]u_L, & |\varpi(t)| \geq u_L \\ \varpi(t), & |\varpi(t)| < u_L \end{cases} \tag{2}$$

where $u_L$ is the boundary of $u(t)$. Obviously, when $|\varpi(t)| = u_L$, there is a sharp angle between the curve of the control input $\varpi(t)$ and the applied control $u(t)$, resulting in the backstepping method not being applied directly, such that the subsequent smooth function can approximately represent the system's saturation

$$I_1 = u_L \times \tanh(p) = \frac{u_L(e^p - e^{-p})}{e^p + e^{-p}} \tag{3}$$

where $p = \varpi/u_L$; then, the saturation input $sat[\varpi(t)]$ in (2) further establishes that

$$sat[\varpi(t)] = I_1 + I_2 = u_L \times \tanh(p) + I_2 \tag{4}$$

where $I_2 = sat[\varpi(t)] - I_1$ is a bounded function, and the bound is straightforward to verify that

$$|I_2| = |sat[\varpi(t)] - I_1| \leq u_L(1 - \tanh(1)) \tag{5}$$

where $u_L(1 - \tanh(1)) = A$ and $A > 0$.

Notice that $I_2$ increases from 0 to $A$ as $|\varpi|$ changes from 0 to $u_L$; when $|\varpi|$ is beyond range, the value of $I_2$ decreases from $A$ to 0 as $|\varpi|$ changes.

To facilitate further study, the following assumptions and partial lemmas are given:

**Assumption 1.** *There exists a constant $\tau > 0$ such that the auxiliary control signal $\left|\tilde{\Gamma}\right| \le \tau$.*

**Assumption 2** ([36]). *The reference signal $y_d$ and its first-order derivatives $\dot{y}_d$ are continuous and bounded. There exist positive constants $B_0$, $\underline{B}_0$, $\bar{B}_0$, and $B_1$, which satisfy $\max\{\underline{B}_0, \bar{B}_0\} \le B_0$, and for $\forall t \ge 0$, $-\underline{B}_0 \le y_d(t) \le \bar{B}_0$, $|\dot{y}_d(t)| \le B_1$.*

**Remark 1.** *The traditional backstepping method [37–41] is a popular approach for trajectory tracking and control of nonlinear systems. However, it requires knowledge of the nth-order derivative of the reference signal, which may not be available or practical in some specific industrial fields, such as robotics, aerospace, and transportation. For example, in aerospace, the reference trajectory of an aircraft or a spacecraft may be preplanned or provided by a ground station, but the availability and accuracy of the nth-order derivative of the reference signal may be limited by various factors, such as atmospheric disturbances, sensor noise, etc. The implementation of the method proposed in this paper is made simpler, and the computational burden is decreased, since there is no need for information about the higher-order derivatives of the reference signal.*

**Lemma 1** ([16]). *There are two variables, $F > 0$ and $G \in \mathbb{R}$, and the property listed below applies to the hyperbolic function $\tanh(\cdot)$.*

$$0 \le |G| - G\tanh\left(\frac{G}{F}\right) \le 0.2785F \tag{6}$$

**Lemma 2.** *For $\forall a \in \mathbb{R}$ and $F > 0$, the hyperbolic function $\tanh(\cdot)$ has unique properties.*

$$-a\tanh\left(\frac{a}{F}\right) \le 0 \tag{7}$$

Our control goal is to developed a low-computation adaptive self-triggered controller such that the tracking error is as small as desired and the output of system (1) can track the reference signal $y_d$ effectively.

### 2.2. RBF NNs Approximation Design

To achieve the control objective, we apply the RBF NNs' approximation capability to handle the unknown nonlinear function $\Psi(\mathcal{X})$: $\Omega_{\mathcal{X}} \subset \mathbb{R}$ on a compact set $\Omega_{\mathcal{X}} \subset \mathbb{R}^d$ with arbitrary accuracy [42–45]; it follows that

$$\Psi(\mathcal{X}) = W^T\phi(\mathcal{X}) + \varepsilon(\mathcal{X}) \tag{8}$$

where $\mathcal{X} \in \Omega_{\mathcal{X}} \subset \mathbb{R}^d$ and $W = (W_1, W_2, \ldots, W_d) \subset \mathbb{R}^L$ represent the input vector, with $d \ge 1$ denoting ideal weight vectors; $L$ is RBF NNs' node number satisfying $L > 1$. The basis function vector $\phi(\mathcal{X})$ is expanded as $[\phi_1(\mathcal{X}_1), \ldots, \phi_L(\mathcal{X}_d)] \in \mathbb{R}^L$, where the Gaussian function $\phi_i(\mathcal{X})$ is expressed as

$$\phi_i(\mathcal{X}) = \exp\left(\frac{-(\mathcal{X} - \zeta_i)^T(\mathcal{X} - \zeta_i)}{\Xi_i^2}\right), 1 \le i \le L \tag{9}$$

where $\zeta_i \in \mathbb{R}^\delta$ denotes the center of basis functions, and its width is expressed by $\Xi_i$. The approximation error $\varepsilon(\mathcal{X})$ satisfies $|\varepsilon(\mathcal{X})| = |\Psi(\mathcal{X}) - \Psi(\mathcal{X}|W)| \le \bar{\varepsilon}$, and $\bar{\varepsilon} > 0$ is a constant. $\Psi(\mathcal{X}|W)$ stands for the actual value of the unknown continuous function of the system under consideration.

## 3. Controller Design

In this section, a controller that is structurally and computationally efficient is presented. Define the following coordinate transformation:

$$e_1 = y - y_d \tag{10}$$

$$e_i = x_i - \alpha_{i-1}, i = 2, \ldots, n-1 \tag{11}$$

$$e_n = x_n - \alpha_{n-1} - \tilde{\Gamma} \tag{12}$$

where $e_1$, $e_i$, and $y_d$ are the output tracking error, the difference between virtual control signal $\alpha_{i-1}$ and state $x_i$, and the reference signal, respectively. $\dot{\tilde{\Gamma}} = -\tilde{\Gamma} + I_1 - v_{std}$ is an auxiliary control signal which is given later. Then, in accordance with the requirements, we design two novel error transformation functions:

$$\xi_i = \cos^2(\frac{\pi e_i}{2\sigma_i}), i = 1, \ldots, n \tag{13}$$

$$\eta_i = \tan(\frac{\pi e_i}{2\sigma_i}), i = 1, \ldots, n \tag{14}$$

where $\sigma_i$ denotes the upper bound of $|e_i|$, which is provided later in the following design. The adaptive law and virtual controller are given as follows:

$$\alpha_i = -c_i \eta_i - \hat{W}_i^T \phi_i(\bar{x}_i) \tag{15}$$

$$\dot{\hat{W}}_i = -\hat{W}_i + \frac{l_i \eta_i}{\sigma_i \xi_i} \phi_i(\bar{x}_i), \ i = 1, 2, \ldots, n \tag{16}$$

where $l_i > 0$ and $c_i > 0$ are design parameters. $-c_i \eta_i$ denotes the design constraint processing scheme, which aims to restrict the error variable $e_i$ in $(-\sigma_i, \sigma_i)$. To improve the system performance, we choose [46]

$$\sigma_i = (\sigma_{i0} - \sigma_{i\infty})e^{-N_i t} + \sigma_{i\infty}, i = 1, \ldots, n \tag{17}$$

where $\sigma_{i\infty} > 0$ and $N_i > 0$ are freely selectable design parameters. They represent the limit value of $|e_i|$ and the convergence rate, respectively. Moreover, the selection of $\sigma_{i0}$ needs to have the following conditions:

$$|e_i(0)| < \sigma_{i0}, i = 1, \ldots, n \tag{18}$$

The realization of $|e_i(t)| < \sigma_i(t), t \geq 0$ depends on this circumstance.

## 4. Stability Analysis

In a closed-loop system, all command signals should be bounded under sufficient conditions, as demonstrated by the following lemma:

**Lemma 3.** By the boundedness of $e_i$, $\dot{e}_i$, $\eta_i$, and $\dot{x}_1, \ldots, \dot{x}_i$ ($i = 1, \ldots, n$), we can deduce that $\dot{\alpha}_i$ is bounded.

**Proof.** From (14) and (15), we can obtain $\dot{\alpha}_i$ and $\dot{\eta}_i (i = 1, \ldots, n)$ as

$$\dot{\alpha}_i = -c_i \dot{\eta}_i - \dot{\hat{W}}_i^T \phi_i(\bar{x}_i) - \hat{W}_i^T \frac{\partial \phi_i}{\partial \bar{x}_i} \dot{\bar{x}}_i \tag{19}$$

$$\dot{\eta}_i = \frac{\pi}{2} \frac{\dot{e}_i \sigma_i - e_i \dot{\sigma}_i}{\sigma_i^2 \xi_i} \tag{20}$$

According to the design of RBF NNs, it is easy to obtain $\|\phi_i(\bar{x}_i)\| \in \mathcal{L}^\infty$ and $\|\partial\phi_i/\partial\bar{x}_i\| \in \mathcal{L}^\infty$. It is noted by the properties of $\xi_i$ and $\eta_i$ that $\xi^{-1} \in \mathcal{L}^\infty$ and $\eta_i \in \mathcal{L}^\infty$ are equivalent. By reviewing (17), one has

$$\sigma_{i\infty} < \sigma_i < \sigma_{i0}, \sigma_{i0}^{-1} < \sigma_i^{-1} < \sigma_{i\infty}^{-1}$$
$$N_i(\sigma_{i\infty} - \sigma_{i0}) < \dot{\sigma}_i < 0, i = 1, \dots, n \tag{21}$$

From (19)–(21), we can obtain that if $\xi_i^{-1}$, $e_i$, and $\dot{e}_i$ are bounded, then the boundedness of $\dot{\eta}_i$ can be guaranteed. In light of (16), by adding stable bounded inputs and outputs, we can conclude that $\|\hat{W}_i\|$ and $\|\dot{\hat{W}}_i\|$ can remain bounded when $\xi_i^{-1}$ is bounded. Thus, the proof is complete. $\square$

Next, we can conclude the following:

**Theorem 1.** *Considering a class of strict-feedback system (1) with saturated inputs, given initial conditions (18), with the support of Lemmas 1 and 2 and Assumptions 1 and 2, the following theory can be established:*

*(1)   The output tracking error gradually approaches and stabilizes within the residual set $(-\sigma_i, \sigma_i)$ as time progresses.*
*(2)   The boundedness of all signals in a closed-loop system is guaranteed.*
*(3)   The Zeno phenomena are successfully avoided.*

**Proof.**  In the beginning, the closed-loop dynamics are formulated. Define $\alpha_0 = y_d$. Combining (1), (10), and (11), the first-order derivative of $e_i$ ($i = 1, 2, \dots, n-1$) is obtained as follows, where the $n$th step is redesigned due to the inclusion of the self-triggered mechanism and the saturation input.

$$\dot{e}_i = \dot{x}_i - \dot{\alpha}_{i-1}$$
$$= e_{i+1} - \dot{\alpha}_{i-1} + \Phi_i(\bar{x}_i) - c_i\eta_i - \hat{W}_i^T\phi_i(\bar{x}_i) \tag{22}$$

The Lyapunov function candidate is selected as $V_i = V_{i1} + V_{i2}$, where

$$V_{i1} = \frac{1}{\pi}\eta_i^2, V_{i2} = \frac{1}{2l_i}\tilde{W}_i^T\tilde{W}_i, (i = 1, \dots, n) \tag{23}$$

where $\tilde{W}_i = W_i - \hat{W}_i$, and $\hat{W}_i$ denotes the estimation of $W_i$.

The derivative of $V_{i1}$ can be found by (20) and (22) as

$$\dot{V}_{i1} = \frac{\eta_i}{\xi_i\sigma_i}(\Phi_i(\bar{x}_i) - c_i\eta_i - \hat{W}_i^T\phi_i(\bar{x}_i)$$
$$+ e_{i+1} - \dot{\alpha}_{i-1} - \frac{e_i\dot{\sigma}_i}{\sigma_i}), i = 1, 2, \dots, n-1 \tag{24}$$

Then, we need to prove by the converse method that all error variables are constrained within the predetermined set [47], i.e., when $t \geq 0$, they satisfy

$$|e_i(t)| < \sigma_i(t), i = 1, \dots, n. \tag{25}$$

Assume the relationship between at least one error variable and a specific time point exists as follows:

$$|e_q(t_m)| \geq \sigma_q(t_m), q \in \{1, \dots, n\} \tag{26}$$

where $t_m < t_{m+1}$ and $m \in Z^+$; then, define the time at which (25) is first violated as $t_1$. In light of (18), we know that $t_1 > 0$. Thus,

$$|e_i(t)| < \sigma_i(t), i = 1, \ldots, n, t < t_1. \tag{27}$$

$$\lim_{t \to t_1^-} |e_q(t)| = \sigma_q(t_1), q \in \{1, \ldots, n\}. \tag{28}$$

The aforementioned negative circumstance is reversed throughout the subsequent analysis and proof process.　□

**Step 1**: In this step, we first analyze the dynamical behavior of the output tracking error. The boundedness of $y = x_1$ on $[0, t_1)$ is ensured with the support of Assumption 1, (10), and (27). That is, the system state $x_1$ always remains within the compact set $\Omega_1$ when $t < t_1$. Using the approximation capability of RBF NNs, we have

$$\Phi_1(\bar{x}_1) = W_1^T \phi_1(x_1) + \varepsilon_1 \tag{29}$$

where $\varepsilon_1$ is the approximation error satisfying the condition of $|\varepsilon_1| \leq \bar{\varepsilon}_1$.

Bringing (29) to (24) with $i = 1$, $\dot{V}_{11}$ can be deduced

$$
\begin{aligned}
\dot{V}_{11} &= \frac{\eta_1}{\zeta_1 \sigma_1}(W_1^T \phi_1(x_1) + \varepsilon_1 - c_1 \eta_1 - \hat{W}_1^T \phi_1(x_1) + e_2 - \dot{y}_d - \frac{e_1 \dot{\sigma}_1}{\sigma_1}) \\
&= \frac{\eta_1}{\zeta_1 \sigma_1}(\Lambda_1 - c_1 \eta_1 + \tilde{W}_1^T \phi_1(\bar{x}_1)) \\
\Lambda_1 &= \varepsilon_1 + e_2 - \dot{y}_d - \frac{e_1 \dot{\sigma}_1}{\sigma_1}
\end{aligned}
\tag{30}
$$

By analyzing Assumption 2, (21), and (27), it is clear that $\varepsilon_1, e_1, e_2, \dot{y}_d, \dot{\sigma}_1$, and $1/\sigma_1$ are bounded. In summary, when $t < t_1$, $\Lambda_1$ is guaranteed to be bounded. For convenience, we denote $|\Lambda_1| \leq \delta_1, t < t_1$. Therefore, (30) can be rewritten as

$$\dot{V}_{11} \leq \frac{\eta_1}{\zeta_1 \sigma_1} \tilde{W}_1^T \phi_1(x_1) + \frac{\eta_1}{\zeta_1 \sigma_1}(\delta_1 - c_1 \eta_1) \tag{31}$$

$$\leq \frac{\eta_1}{\zeta_1 \sigma_1} \tilde{W}_1^T \phi_1(x_1) + \frac{1}{\zeta_1 \sigma_1}(\delta_1 |\eta_1| - c_1 \eta_1^2) \tag{32}$$

According to Young's inequality and recalling (23), one has

$$\delta_1 |\eta_1| \leq \frac{1}{2c_1} \delta_1^2 + \frac{c_1}{2} \eta_1^2 = \frac{1}{2c_1} \delta_1^2 + \frac{\pi c_1}{2} V_{11} \tag{33}$$

Thus, (31) can be rewritten as

$$\dot{V}_{11} \leq \frac{\eta_1}{\zeta_1 \sigma_1} \tilde{W}_1^T \phi_1(x_1) + \frac{1}{\zeta_1 \sigma_1}(\omega_1 - h_1 V_{11}) \tag{34}$$

where $\omega_1 = \frac{1}{2c_1} \delta_1^2$ and $h_1 = \frac{\pi c_1}{2}$

On the basis of (16) and (23), we can obtain

$$
\begin{aligned}
\dot{V}_{12} &= -\frac{1}{l_1} \tilde{W}_1^T \dot{\hat{W}}_1 = \frac{1}{l_1} \tilde{W}_1^T \hat{W}_1 - \frac{\eta_1}{\zeta_1 \sigma_1} \tilde{W}_1^T \phi_1(x_1) \\
&= \frac{1}{l_1} \tilde{W}_1^T W_1 - \frac{1}{l_1} \tilde{W}_1^T \tilde{W}_1 - \frac{\eta_1}{\zeta_1 \sigma_1} \tilde{W}_1^T \phi_1(x_1)
\end{aligned}
\tag{35}
$$

Using Young's inequality again, it is concluded that

$$\frac{1}{l_1}\tilde{W}_1^T W_1 \leq \frac{1}{2l_1}\tilde{W}_1^T \tilde{W}_1 + \frac{1}{2l_1}W_1^T W_1 \tag{36}$$

Bring the above equation back to (35), $\dot{V}_{12}$ further satisfies

$$\begin{aligned}
\dot{V}_{12} &\leq -\frac{1}{2l_1}\tilde{W}_1^T \tilde{W}_1 + \frac{1}{2l_1}W_1^T W_1 - \frac{\eta_1}{\xi_1\sigma_1}\tilde{W}_1^T \phi_1(x_1) \\
&\leq -V_{12} + \frac{1}{2l_1}W_1^T W_1 - \frac{\eta_1}{\xi_1\sigma_1}\tilde{W}_1^T \phi_1(x_1)
\end{aligned} \tag{37}$$

To sum up, we arrive at

$$\begin{aligned}
\dot{V}_1 &= \dot{V}_{11} + \dot{V}_{12} \\
&\leq \frac{1}{\xi_1\sigma_1}(\omega_1 - h_1 V_{11}) - V_{12} + \frac{1}{2l_1}W_1^T W_1
\end{aligned} \tag{38}$$

Next, the boundedness of $V_1$ is illustrated by discussing two different cases of $V_{11}$.

Case 1: $V_{11} \leq \frac{\omega_1}{h_1} + \lambda_1$, where $\lambda_1 > 0$ is a parameter used for analysis, the definition of which is given subsequently. Apparently,

$$V_{12} = V_1 - V_{11} \geq V_1 - \frac{\omega_1}{h_1} - \lambda_1 \tag{39}$$

According to (13) and (21), term $\frac{1}{\xi_1\sigma_1}(\omega_1 - h_1 V_{11})$ is bounded in this case. So, we have

$$\frac{1}{\xi_1\sigma_1}(\omega_1 - h_1 V_{11}) \leq \frac{\omega_1}{\xi_1\sigma_1} \leq \hbar_1 \tag{40}$$

By putting (39) and (40) into (38), we can obtain

$$\dot{V}_1 \leq -V_1 + \frac{1}{2l_1}W_1^T W_1 + \frac{\omega_1}{h_1} + \lambda_1 + \hbar_1 \tag{41}$$

Case 2: $V_{11} > \frac{\omega_1}{h_1} + \lambda_1$. In this case, there is

$$\omega_1 - h_1 V_{11} < -\lambda_1 h_1 \tag{42}$$

From (14), (13), (21), and (23), $\frac{1}{\xi_1\sigma_1}$ can be deduced as

$$\frac{1}{\xi_1\sigma_1} = \frac{\tan^2(\frac{\pi e_1}{2\sigma_1})}{\sigma_1\sin^2(\frac{\pi e_1}{2\sigma_1})} = \frac{\eta_1^2}{\sigma_1\beta_1^2} = \frac{\pi}{\sigma_1\beta_1^2}V_{11} \geq \frac{\pi}{\sigma_{10}}V_{11} \tag{43}$$

where $\beta_1 = \sin(\frac{\pi e_1}{2\sigma_1})$.

Combining (42) and (43), we can obtain

$$\frac{1}{\xi_1\sigma_1}(\omega_1 - h_1 V_{11}) < -\frac{\pi\lambda_1 h_1}{\sigma_{10}}V_{11} \tag{44}$$

Let $\lambda_1 = \frac{\sigma_{10}}{\pi h_1}$ and earn

$$\frac{1}{\xi_1\sigma_1}(\omega_1 - h_1 V_{11}) < -V_{11} \tag{45}$$

Substituting (45) into (38) yields

$$\dot{V}_1 < -V_1 + \frac{1}{2l_1} W_1^T W_1 \tag{46}$$

Incorporating (41) and (46), $\dot{V}_1$ can be modified as

$$\begin{aligned}
\dot{V}_1 &< -(V_{11} + V_{12}) + \frac{1}{2l_1} W_1^T W_1 \\
&< -V_1 + \gamma_1 \\
\gamma_1 &= \frac{1}{2l_1} W_1^T W_1 + \frac{\omega_1}{h_1} + \lambda_1 + \hbar_1
\end{aligned} \tag{47}$$

By integrating over both sides of (47), it is not difficult to see that

$$V_1 < (V_1(0) - \gamma_1)e^{-t} + \gamma_1, t < t_1 \tag{48}$$

Step i ($2 \leq i \leq n-1$): The dynamic behavior of $e_i$ is described in this step. Firstly, it is necessary to validate that the system state $\bar{x}_i$ is guaranteed at the set $\Omega_i$ in $[0, t_1)$. The time interval $[0, t_1)$ forms the basis of the analysis as follows:

(1)  According to (23) and (48), it can be proved that $\eta_{i-1}$ and $\|\hat{W}_{i-1}\|$ are bounded.
(2)  Under (15), the boundedness of $\alpha_{i-1}$ can be derived directly.
(3)  For step $i$, using (11) and (27), we can prove that $x_i$ is bounded.

Recall that establishing the boundedness of $x_1$ in step 1 yields that the system states $x_1, \ldots, x_i$ are guaranteed in the compact space $\Omega_i$, and the unknown nonlinear factor $\Phi_i(\bar{x}_i)(2 \leq i \leq n-1)$ in (24) is approximated by RBF NNs, i.e.,

$$\Phi_i(\bar{x}_i) = W_i^T \phi_i(\bar{x}_i) + \varepsilon_i \tag{49}$$

where $\varepsilon_i$ with $|\varepsilon_i| \leq \bar{\varepsilon}_i$ is the approximation error.

Bringing (49) to (24), we have

$$\begin{aligned}
\dot{V}_{i1} &= \frac{\eta_i}{\zeta_i \sigma_i} \left( W_i^T \phi_i(\bar{x}_i) + \varepsilon_i - c_i \eta_i - \hat{W}_i^T \phi_i(\bar{x}_i) + e_{i+1} - \dot{\alpha}_{i-1} - \frac{e_i \dot{\sigma}_i}{\sigma_i} \right) \\
&= \frac{\eta_i}{\zeta_i \sigma_i} \left( \Lambda_i - c_i \eta_i + \tilde{W}_i^T \phi_i(\bar{x}_i) \right) \\
\Lambda_i &= \varepsilon_i + e_{i+1} - \dot{\alpha}_{i-1} - \frac{e_i \dot{\sigma}_i}{\sigma_i}
\end{aligned} \tag{50}$$

Notice from (21) and (27) that $1/\sigma_i, \dot{\sigma}_i, e_1, e_i$, and $e_{i+1}$ are bounded as $t < t_1$. Then, we ensure that $\dot{\alpha}_{i-1}$ is bounded on $[0, t_1)$. By $\Phi_{i-1}$ being continuous and $x_{i-1}$ being bounded, we can obtain that for $[0, t_1)$, the nonlinear function $\Phi_{i-1}(\bar{x}_{i-1})$ has a bound. On this basis, the boundedness of $x_{i-i}$ can also be guaranteed, as can $\dot{x}_{i-1}$ governed by (1). Based on Lemma 3, $\dot{\alpha}_{n-1}$ is also bounded on the interval $[0, t_1)$.

The results presented above support the existence of a positive constant $\delta_i$, which makes $|\Lambda_i| < \delta_i, t < t_1$. Thus, (50) becomes

$$\dot{V}_{i1} \leq \frac{\eta_i}{\zeta_i \sigma_i} \tilde{W}_i^T \phi_i(\bar{x}_i) + \frac{\eta_i}{\zeta_i \sigma_i}(\delta_i - c_i \eta_i) \tag{51}$$

$$\leq \frac{\eta_i}{\zeta_i \sigma_i} \tilde{W}_i^T \phi_i(\bar{x}_i) + \frac{1}{\zeta_i \sigma_i}(\delta_i |\eta_i| - c_i \eta_i^2) \tag{52}$$

Using Young's inequality and noticing $V_{i1} = \frac{1}{\pi}\eta_i^2$, we have

$$\delta_i|\eta_i| \le \frac{1}{2c_i}\delta_i^2 + \frac{c_i}{2}\eta_i^2 = \frac{1}{2c_i}\delta_i^2 + \frac{\pi c_i}{2}V_{i1} \tag{53}$$

Then, (51) turns into

$$\dot{V}_{i1} \le \frac{\eta_i}{\xi_i\sigma_i}\tilde{W}_i^T\phi_i(\bar{x}_i) + \frac{1}{\xi_i\sigma_i}(\omega_i - h_iV_{i1}) \tag{54}$$

where $\omega_i = \frac{1}{2c_i}\delta_i^2$ and $h_i = \frac{\pi c_i}{2}$.

With the support of (16) and (23), the derivative of $V_{i2}$ is

$$\begin{aligned}\dot{V}_{i2} &= -\frac{1}{l_i}\tilde{W}_i^T\dot{\hat{W}}_i = \frac{1}{l_i}\tilde{W}_i^T\hat{W}_i - \frac{\eta_i}{\xi_i\sigma_i}\tilde{W}_i^T\phi_i(\bar{x}_i)\\ &= \frac{1}{l_i}\tilde{W}_i^TW_i - \frac{1}{l_i}\tilde{W}_i^T\tilde{W}_i - \frac{\eta_i}{\xi_i\sigma_i}\tilde{W}_i^T\phi_i(\bar{x}_i)\end{aligned} \tag{55}$$

Applying the same method as (36), we arrive at

$$\frac{1}{l_i}\tilde{W}_i^TW_i \le \frac{1}{2l_i}W_i^TW_i + \frac{1}{2l_i}\tilde{W}_i^T\tilde{W}_i \tag{56}$$

Substituting (56) into (55) yields that $\dot{V}_{i2}$ further satisfies

$$\begin{aligned}\dot{V}_{i2} &\le \frac{1}{2l_i}W_i^TW_i - \frac{1}{2l_i}\tilde{W}_i^T\tilde{W}_i - \frac{\eta_i}{\xi_i\sigma_i}\tilde{W}_i^T\phi_i(\bar{x}_i)\\ &\le -V_{i2} + \frac{1}{2l_i}W_i^TW_i - \frac{\eta_i}{\xi_i\sigma_i}\tilde{W}_i^T\phi_i(\bar{x}_i)\end{aligned} \tag{57}$$

Integrate (54) and (57), the derivative of $V_i$ can be further modified as

$$\begin{aligned}\dot{V}_i &= \dot{V}_{i1} + \dot{V}_{i2}\\ &\le \frac{1}{\xi_i\sigma_i}(\omega_i - h_iV_{i1}) - V_{i2} + \frac{1}{2l_i}W_i^TW_i\end{aligned} \tag{58}$$

In this step, we also elaborate the boundedness of $V_1$ by analyzing two different cases of $V_{i1}$.

Case 1: $V_{i1} \le \frac{\omega_i}{h_i} + \lambda_i$, where $\lambda_i > 0$ is an analysis parameter. Obviously,

$$V_{i2} = V_i - V_{i1} \ge V_i - \frac{\omega_i}{h_i} - \lambda_i \tag{59}$$

From (13) and (21), it is straightforward to see that

$$\frac{1}{\xi_i\sigma_i}(\omega_i - h_iV_{i1}) \le \frac{\omega_i}{\xi_i\sigma_i} \le \hbar_i \tag{60}$$

Inserting (59) and (60) into (58), we arrive at

$$\dot{V}_i \le -V_i + \frac{1}{2l_i}W_i^TW_i + \frac{\omega_i}{h_i} + \lambda_i + \hbar_i \tag{61}$$

Case 2: $V_{i1} > \frac{\omega_i}{h_i} + \lambda_i$. In this case, there are

$$\omega_i - h_iV_{i1} < -\lambda_ih_i \tag{62}$$

Recalling (14), (13), (21), and (23), define $\beta_i = \sin(\frac{\pi e_i}{2\sigma_i})$, thus obtaining

$$\frac{1}{\xi_i \sigma_i} = \frac{\tan^2(\frac{\pi e_i}{2\sigma_i})}{\sigma_i \sin^2(\frac{\pi e_i}{2\sigma_i})} = \frac{\eta_i^2}{\sigma_i \beta_i^2} = \frac{\pi}{\sigma_i \beta_i^2} V_{i1} \geq \frac{\pi}{\sigma_i} V_{i1} \geq \frac{\pi}{\sigma_{i0}} V_{i1} \tag{63}$$

Sorting (62) and (63), and taking $\lambda_i = \frac{\sigma_{i0}}{\pi h_i}$, we can derive

$$\frac{1}{\xi_i \sigma_i}(\omega_i - h_i V_{i1}) < -V_{i1} \tag{64}$$

Bringing (64) into (58) yields that we arrive at

$$\dot{V}_i < -(V_{i1} + V_{i2}) + \frac{1}{2l_i} W_i^T W_i$$
$$< -V_i + \frac{1}{2l_i} W_i^T W_i \tag{65}$$

Merging (61) and (65), there are

$$\dot{V}_i < -V_i + \gamma_i$$
$$\gamma_i = \frac{1}{2l_i} W_i^T W_i + \frac{\omega_i}{h_i} + \lambda_i + \hbar_i \tag{66}$$

Integrating both ends of the above equation, when $t < t_1$, it can be clearly seen that

$$V_i < (V_i(0) - \gamma_i)e^{-t} + \gamma_i \tag{67}$$

Step n: Incorporating (12) into (1) yields that $\dot{e}_n$ is computed as

$$\dot{e}_n = \dot{x}_n - \dot{\alpha}_{n-1} - \dot{\tilde{\Gamma}}$$
$$= u(\varpi) + \Phi_n(\bar{x}_n) - \dot{\alpha}_{n-1} - \dot{\tilde{\Gamma}} \tag{68}$$

where $\dot{\tilde{\Gamma}} = -\tilde{\Gamma} + I_1 - v_{std}$ is the defined dynamic system [32].

Taking $V_{n1}$ in the Lyapunov function (23) and combining it with (20), for $\dot{V}_{n1}$, it is straightforward to observe that

$$\dot{V}_{n1} = \frac{1}{\pi} \eta_n \dot{\eta}_n$$
$$= \frac{\eta_n}{\sigma_n \xi_n}\left(u(\varpi) + \Phi_n(\bar{x}_n) - \dot{\alpha}_{n-1} - \dot{\tilde{\Gamma}} - \frac{e_n \dot{\sigma}_n}{\sigma_n}\right) \tag{69}$$

In a similar way to the previous step, it can be recursively deduced that the system states $x_1, \ldots, x_i, \ldots, x_n$ guarantee in a compact space $\Omega_n$. Therefore, the unknown function $\Phi_n(\bar{x}_n)$ is approximated via RBF NNs, i.e.,

$$\Phi_n(\bar{x}_n) = W_n^T \phi_n(\bar{x}_n) + \varepsilon_n \tag{70}$$

where $\varepsilon_n$ is the approximation error and satisfies $|\varepsilon_n| \leq \bar{\varepsilon}_n$.

Substituting (70) into (69) and recalling (2) and (4), one has

$$\dot{V}_{n1} = \frac{\eta_n}{\sigma_n \xi_n}\left(I_2 + W_n^T \phi_n(\bar{x}_n) + \varepsilon_n - \dot{\alpha}_{n-1} \right.$$
$$\left. + \tilde{\Gamma} + v_{std} - \frac{e_n \dot{\sigma}_n}{\sigma_n}\right) \tag{71}$$

The self-triggered mechanism is designed as

$$v_{std}(t) = \chi(t_\iota), \quad \forall t \in [t_\iota, t_{\iota+1})$$

$$t_{\iota+1} = t_\iota + \frac{k_\sigma |v_{std}(t)| + k_D}{\max\{|\bar{\chi}(t)|, k_c\}} \tag{72}$$

where $t_\iota, t_{\iota+1} \in Z^+$, $0 < k_\sigma < 1$. $k_D$ and $k_c$ are positive design parameters. $k_\sigma |v_{std}(t)| + k_D$ represents the interval between two successfully triggered control signals; $|\bar{\chi}(t)|$ with $\bar{\chi}(t) = \dot{\chi}(t)|_{t=t_\iota}$ and $k_c$ denote the change rates of the control signal. When (72) is triggered, $v_{std}(t) = \chi(t_\iota)$ will be input immediately into the system (1). The following trigger point $t_{\iota+1}$ will also be obtained at the same time and control signal $v_{std}(t)$ will remain at $\chi(t_\iota)$ during $t \in [t_\iota, t_{\iota+1})$. $\chi(t)$ is expressed as

$$\chi(t) = -(1 + k_\sigma)[\alpha_n \tanh(\frac{\mathrm{K}\alpha_n}{\mathrm{P}}) + k_m \tanh(\frac{\mathrm{K}k_m}{\mathrm{P}})] \tag{73}$$

where $\mathrm{K} = \eta_n / (\sigma_n \xi_n)$, $\mathrm{P}$ and $k_m > \frac{k_D}{1-k_\sigma}$ are positive design parameters.

**Remark 2.** *Given that the next trigger point is calculated, we introduce a term* $-(1 + k_\sigma)k_m$ $\tanh(\frac{k_m \mathrm{K}}{\mathrm{P}})$ *to mitigate potential calculation errors. This compensation method, which is widely used in nonlinear system control, has proven effective.*

**Remark 3.** *In contrast to the conventional event-triggered mechanism [26,27,31,48] which requires continuous monitoring of thresholds, the self-triggered scheme proposed in this paper uses current system state information to determine the next trigger point for controller updates. This approach overcomes the monitoring challenge and maintains the communication resource-saving benefits of the event-triggered mechanism.*

From (72), $|v_{std}(t_{\iota+1}) - v_{std}(t)| \leq k_\sigma |v_{std}(t)| + k_D$ is derived. Additionally, we then obtain $|\chi(t) - v_{std}(t_\iota)| \leq k_\sigma |v_{std}(t)| + k_D$. By setting the time-varying continuous function $\rho_1(t_\iota) = \rho_2(t_\iota) = 0$, $\rho_1(t_{\iota+1}) = \rho_2(t_{\iota+1}) = \pm 1$ and $|\rho_1(t_\iota)| \leq 1$, $|\rho_2(t_\iota)| \leq 1$, $\forall t \in [t_\iota, t_{\iota+1})$, $(1 + \rho_1(t)k_\sigma)v_{std}(t) = \chi(t) - \rho_2(t)k_D$ can be obtained. Thus, we have $v_{std}(t) = \frac{\chi(t) - \rho_2(t)k_D}{1 + \rho_1(t)k_\sigma}$.

Since

$$\begin{aligned}
\mathrm{K}v_{std}(t) &= \frac{\mathrm{K}\chi(t)}{1 + \rho_1(t)k_\sigma} - \frac{\mathrm{K}\rho_2(t)k_D}{1 + \rho_1(t)k_\sigma} \\
&\leq -\mathrm{K}\alpha_n \tanh(\frac{\mathrm{K}\alpha_n}{\mathrm{P}}) - \mathrm{K}k_m \tanh(\frac{\mathrm{K}k_m}{\mathrm{P}}) + \left| \frac{\mathrm{K}\rho_2(t)k_D}{1 + \rho_1(t)k_\sigma} \right| \\
&\leq \mathrm{K}\alpha_n - |\mathrm{K}k_m| + \left| \frac{\mathrm{K}k_D}{1 - k_\sigma} \right| + 0.557\mathrm{P} \\
&\leq \mathrm{K}\alpha_n + 0.557\mathrm{P}
\end{aligned} \tag{74}$$

where $\mathrm{K} = \eta_n / (\sigma_n \xi_n)$. Then, combined with (16), (71) is rewritten as

$$\begin{aligned}
\dot{V}_{n1} &= \frac{\eta_n}{\sigma_n \xi_n} (I_2 + \tilde{W}_n^T \phi_n(\bar{x}_n) + \varepsilon_n - \dot{\alpha}_{n-1} \\
&\quad + \tilde{\Gamma} - c_n \eta_n + 0.557\mathrm{P}\frac{\sigma_n \xi_n}{\eta_n} - \frac{e_n \dot{\sigma}_n}{\sigma_n}) \\
&= \frac{\eta_n}{\sigma_n \xi_n} (\Lambda_n - c_n \eta_n + \tilde{W}_n^T \phi_n(\bar{x}_n)) \\
\Lambda_n &= I_2 + \varepsilon_n - \dot{\alpha}_{n-1} + \tilde{\Gamma} + 0.557\mathrm{P}\frac{\sigma_n \xi_n}{\eta_n} - \frac{e_n \dot{\sigma}_n}{\sigma_n}
\end{aligned} \tag{75}$$

where $\tilde{\Gamma}$ is bounded [34]. Based on (5), (14), and (13), $I_2$, $\frac{1}{\eta_n}$, and $\xi_n$ are bounded, so that $|\Lambda_n| < \delta_n, t < t_1$. Through some algebraic manipulators, for $\dot{V}_{n1}$, it can be obtained that

$$\dot{V}_{n1} \leq \frac{\eta_n}{\sigma_n \xi_n} \tilde{W}_n^T \phi_n(\bar{x}_n) + \frac{1}{\sigma_n \xi_n}(\omega_n - h_n V_{n1})$$

$$\dot{V}_{n2} \leq -V_{n2} + \frac{1}{2l_n} W_n^T W_n - \frac{\eta_n}{\sigma_n \xi_n} \tilde{W}_n^T \phi_n(\bar{x}_n)$$

where $h_n = \frac{\pi}{2} c_n$ and $\omega_n = \frac{1}{2c_n} \delta_n^2$. Accordingly, the bound of $\dot{V}_n$ further satisfies

$$\dot{V}_n \leq -V_{n2} + \frac{1}{2l_n} W_n^T W_n + \frac{1}{\sigma_n \xi_n}(\omega_n - h_n V_{n1})$$

It is possible to demonstrate by classified discussion that

$$\dot{V}_n < -V_n + \gamma_n$$

$$\gamma_n = \frac{1}{2l_n} W_n^T W_n + \frac{\omega_n}{h_n} + \lambda_n + \hbar_n$$

where $\lambda_n = \frac{\sigma_{n0}}{\pi h_n}$. In this scenario

$$V_n < (V_n(0) - \gamma_n)e^{-t} + \gamma_n, \ t < t_1 \tag{76}$$

Noticing that $V_{i1} = \frac{1}{\pi} \eta_i^2 = V_i - V_{i2} \leq V_i, \ i = 1, \dots, n$. Combining the results in (48), (67), and (76), and considering the initial condition $V_i(0)$ is bounded, we can see that

$$\eta_i^2 < \pi V_i(0), \ t < t_1$$

It is obvious that $\eta_1, \dots, \eta_n$ remains bounded on the interval $[0, t_1)$. In the light of the definition of $\eta_i$ in (14), it can be stated that for each error variable $e_i$, $|e_i|$ guarantees within the prescribed boundary function $\sigma_i(t), i = 1, \dots, n$. The conclusion (27) under assumption (26) is in contradiction. Therefore, the assumption (26) is not reasonable, thus justifying the conclusion (25).

Invoking (10) and combining (17) and (25) for $i = 1$, it follows that

$$\lim_{t \to \infty} |y(t) - y_d(t)| < \sigma_{1\infty} \tag{77}$$

which demonstrates that the tracking error gradually diminishes until it reaches a residual set in close proximity to 0. Based on (25) and using the same steps as before, it follows by step-by-step derivation that the results of (48), (67), and (76) hold for all $t \geq 0$, i.e.,

$$\dot{V}_i < -V_i + \gamma_i, \ i = 1, \dots, n$$

Further derived

$$V_i = V_{i1} + V_{i2} < (V_i(0) - \gamma_i)e^{-t} + \gamma_i$$

With the aid of (23), we obtain $\eta_i \in \mathcal{L}^\infty$ and $\|\hat{W}_i\| \in \mathcal{L}^\infty, i = 1, \dots, n$. By (15), we obtain $\alpha_i \in \mathcal{L}^\infty$. From (11), (12), and (25), $x_i \in \mathcal{L}^\infty, i = 1, \dots, n$ hold. Up to this point, the boundedness of all signals in the closed-loop system is guaranteed.

Given that $v_{std}$ is bounded, we can infer from (72) that $(k_\sigma |v_{std}(t)| + k_D)/(\max\{|\bar{\chi}(t)|, k_c\})$ is also bounded. Consequently, we can determine $t^* = t_{t+1} - t_t > 0$, where $t^*$ is a bounded minimum time interval, which implies that there will be no Zeno phenomenon [49]. Thus, Theorem 1 can be established.

## 5. Simulation Example and Analysis

### 5.1. Example Model 1

In this section, the effectiveness of our proposed method is verified by a numerical example. Consider the following nonlinear system with input saturation as

$$\begin{cases} \dot{x}_1 = \Phi_1(x_1) + x_2 \\ \dot{x}_2 = \Phi_2(\bar{x}_2) + u(\varpi) \\ y = x_1 \end{cases}$$

where the unknown nonlinear functions $\Phi_1(x_1)$ and $\Phi_2(\bar{x}_2)$ are chosen as $\Phi_1(x_1) = -\frac{1}{3}x_1^3 + x_1 + 0.74\cos(t)$ and $\Phi_2(\bar{x}_2) = 0.1(x_1 + 0.7 - 0.8x_2)$, respectively. The reference trajectory is selected as $y_d = \sin(t)$. Additionally, input $u(\varpi)$ is described by

$$u(\varpi(t)) = sat(\varpi(t)) = \begin{cases} sign(\varpi(t))u_L, & |\varpi(t)| \geq u_L \\ \varpi(t), & |\varpi(t)| < u_L \end{cases}$$

where $u_L = 4$. According to (9), the Gaussian basis functions $\phi_{i,q}(\bar{X}_i)$ of the RBF NNs can be defined as

$$\phi_{i,q}(\bar{X}_i) = e^{-\frac{(\bar{X}_i - \zeta_{i,q})^T(\bar{X}_i - \zeta_{i,q})}{\Xi_{i,q}^2}}, 1 \leq q \leq 5, i = 1, 2$$

where $\bar{X}_1 = X_1$, $\bar{X}_2 = [X_1, X_2]^T$, $\Xi_{i,q} = 2$, $\zeta_{1,q} = 2 + q$, $\zeta_{2,q} = [2 + q, 2 - q]^T$. The parameters of the self-triggered mechanism are set as $k_\sigma = 0.02$, $k_D = 0.12$, $k_c = 0.5$. The designed controller, adaptive law, and auxiliary system are

$$\alpha_1 = -c_1\eta_1 - \hat{W}_1^T\phi_1(\bar{x}_1)$$

$$\alpha_2 = -c_2\eta_2 - \hat{W}_2^T\phi_2(\bar{x}_2)$$

$$\dot{\hat{W}}_1 = -\hat{W}_1 + \frac{l_1\eta_1}{\sigma_1\bar{\zeta}_1}\phi_1(\bar{x}_1)$$

$$\dot{\hat{W}}_2 = -\hat{W}_2 + \frac{l_2\eta_2}{\sigma_2\bar{\zeta}_2}\phi_2(\bar{x}_2)$$

$$\dot{\tilde{\Gamma}} = -\tilde{\Gamma} + I_1 - v_{std}$$

where $c_1 = 1$, $c_2 = 10$, $l_1 = 5$, and $l_2 = 5$. The initial conditions for all variables are chosen as follows: $x_1(0) = 0.07$, $x_2(0) = 0.08$, $\hat{W}_1(0) = 0.5$, $\hat{W}_2(0) = 0.4$, and $\tilde{\Gamma}(0) = 0.3$. With (17) and (18), the performance bounds on $e_1$ and $e_2$ are designated as $\sigma_1 = (0.5 - 0.01)e^{-t} + 0.01$ and $\sigma_2 = (1 - 0.5)e^{-0.5t} + 0.5$, respectively.

The numerical simulation example of the research results in this paper is shown in Figures 1–6, where Figure 1 shows the tracking error and user-specified error boundary, which meet the transient and steady-state tracking behavior. Figure 2 illustrates tracking error trajectories using the command filtering method. The control input trajectory of the system is shown in Figure 3. Figure 4 displays the original input signal and the self-triggered input signal, which demonstrates that the proposed self-triggered controller can effectively conserve communication resources. Figure 5 shows the trajectory of the control input $u$ under the command filter control method for the same control effect. Figure 6 represents the trajectories of adaptive law.

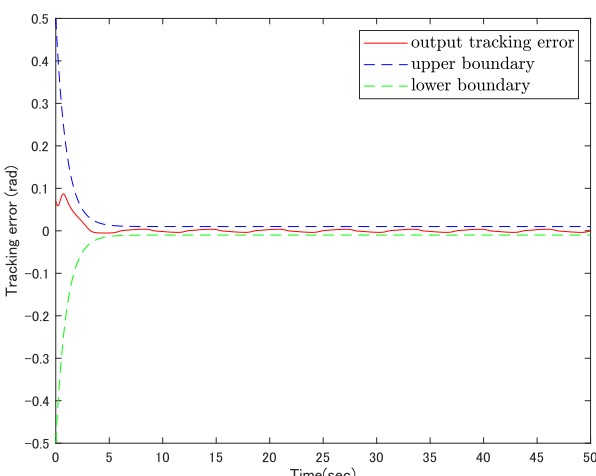

**Figure 1.** Tracking error $y - y_d$ and the error boundaries $\pm\sigma_1$.

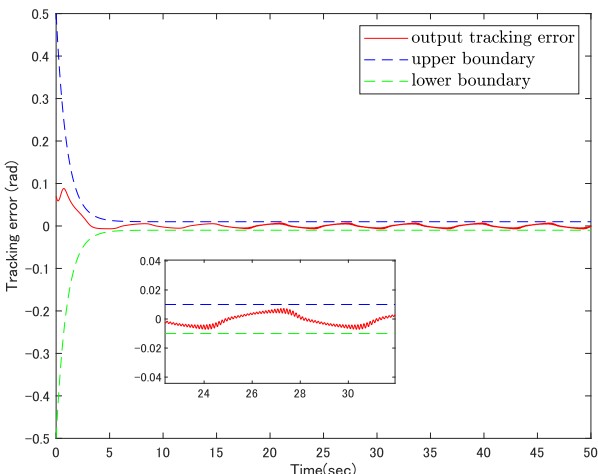

**Figure 2.** Tracking error via the command-filtered control method.

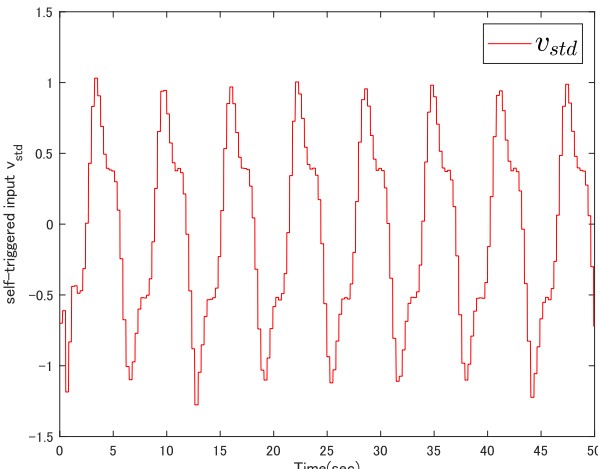

**Figure 3.** Trajectory of self-triggered input $v_{std}$.

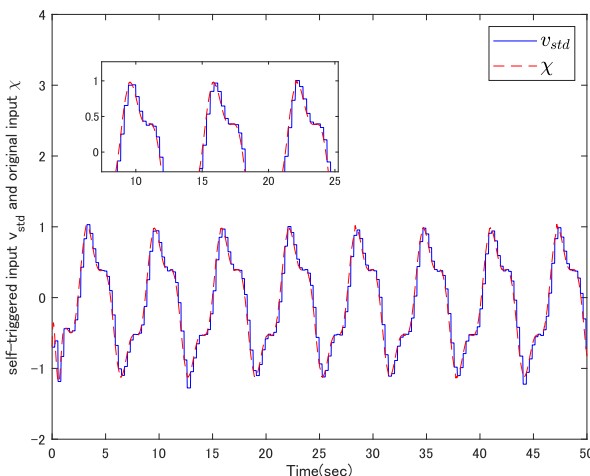

**Figure 4.** Trajectories of self-triggered input $v_{std}$ and original input $\chi$.

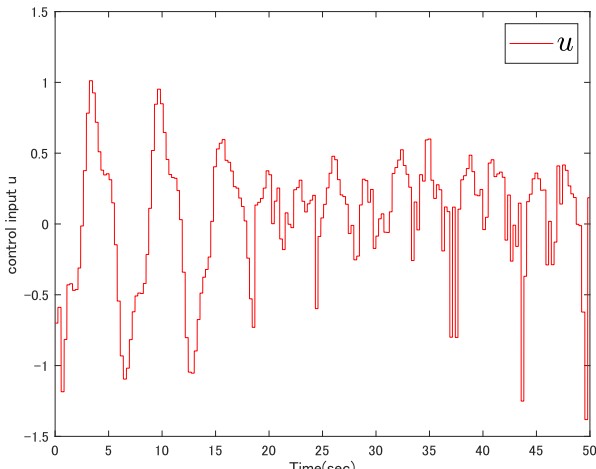

**Figure 5.** Trajectory of control input $v_{std}$ via the command-filtered control method.

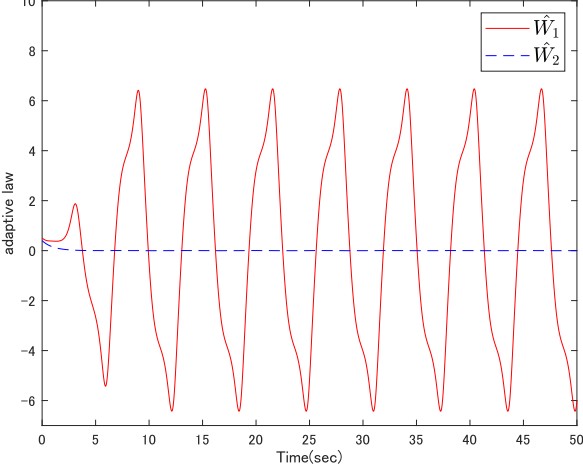

**Figure 6.** Trajectories of adaptive law $\hat{W}_1$ and $\hat{W}_2$.

### 5.2. Example Model 2

Consider a single-link mobile robot arm driven by a brushed DC motor with input saturation, which can be expressed in the following form [50]

$$\mathbb{H}\ddot{p} + \mathbb{F}\dot{p} + \mathbb{Q}\sin(p) = \mathcal{I}$$
$$\mathbb{K}\dot{\mathcal{I}} + \mathbb{D}\mathcal{I} + \mathbb{E}\dot{p} = \mathcal{U}$$

where $q$ represents the link position, and $\mathcal{I}$ and $\mathcal{U}$ represent the motor armature current and input control voltage, respectively. Define $p = x_1$, $\dot{p} = x_2$, $\mathcal{I} = x_3$, and $\mathcal{U} = v_{std}$. Accordingly, the dynamic model can be further shown as

$$\dot{x}_1 = x_2$$
$$\dot{x}_2 = x_3 + \Phi_2(\bar{x}_2)$$
$$\dot{x}_3 = sat(\varpi(t)) + \Phi_3(\bar{x}_3)$$

where the unknown dynamics are $\Phi_2(\bar{x}_2) = -\frac{\mathbb{F}}{\mathbb{H}}x_2 - \frac{\mathbb{Q}}{\mathbb{H}}\sin(x_1)$ and $\Phi_3(\bar{x}_3) = -\frac{\mathbb{D}}{\mathbb{K}}x_3 - \frac{\mathbb{E}}{\mathbb{K}}x_2$, and the actual parameters are given as $\mathbb{H} = 1$, $\mathbb{F} = 1$, $\mathbb{Q} = 10$, $\mathbb{K} = 2.5 \times 10^{-2}$, $\mathbb{D} = 5$, and $\mathbb{E} = 0.9$. Furthermore, the desired reference trajectory is given as $\dot{y}_d = -2y_d + \frac{\pi}{2}$. In addition, the parameters of RBF NNs are chosen in accordance with example 5.1. The parameters of the self-triggered mechanism are set as $k_\sigma = 0.1$, $k_D = 1.2$, and $k_c = 5.5$, and the designed controller and adaptive law are

$$\alpha_1 = -c_1\eta_1$$
$$\alpha_2 = -c_2\eta_2 - \hat{W}_2^T\phi_2(\bar{x}_2)$$
$$\alpha_3 = -c_3\eta_3 - \hat{W}_3^T\phi_3(\bar{x}_3)$$
$$\dot{\hat{W}}_2 = -\hat{W}_2 + \frac{l_2\eta_2}{\sigma_2\zeta_2}\phi_2(\bar{x}_2)$$
$$\dot{\hat{W}}_3 = -\hat{W}_3 + \frac{l_3\eta_3}{\sigma_3\zeta_3}\phi_3(\bar{x}_3)$$

where $c_1 = 1$, $c_2 = 10$, $c_3 = 20$, $l_2 = 5$, and $l_3 = 50$. The initial conditions for all variables are chosen as follows: $x_1(0) = 0.07$, $x_2(0) = x_3(0) = 0.08$, $\hat{W}_2(0) = 0.5$, $\hat{W}_3(0) = 0.4$, $y_d(0) = 0$, and $\tilde{\Gamma}(0) = 0.3$. Next, we define performance functions as $\sigma_1 = (0.5 - 0.01)e^{-t} + 0.01$, $\sigma_2 = (1 - 0.5)e^{-0.5t} + 0.5$, and $\sigma_3 = (2 - 1)e^{-0.5t} + 1$.

The simulation results of the proposed controller being applied to the robot manipulator are shown in Figures 7–10. The error of the link position $p$ with the desired reference trajectory $y_d$ is depicted in Figure 7, in which the transient and steady-state performance guarantees within the boundaries $\pm\sigma_1$ can be seen. Figure 8 shows the trajectory curve of $\mathcal{I}$, and the convergence of the adaptive parameters $W_2$ and $W_3$ are illustrated in Figure 9. The required armature motor voltage and self-triggering inputs are shown in Figure 10. As can be observed, the suggested self-triggering method reduces the controller's update frequency while saving communication resources and increasing the effectiveness of data transmission. Additionally, for the purpose of comparison, applying the command filtering method under the same performance specifications and initial conditions, the tracking error and system input are shown in Figures 11 and 12, respectively. From Figures 10 and 12, it can be seen that the low-computation strategy in this paper requires a control input voltage of 10 Vdc. However, the input voltage required by the command filtering method requires a larger control action.

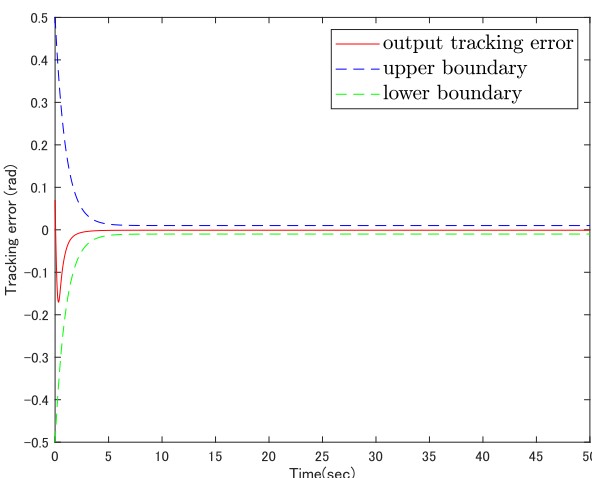

**Figure 7.** Tracking error $q - y_d$ and the error boundaries $\pm \sigma_1$.

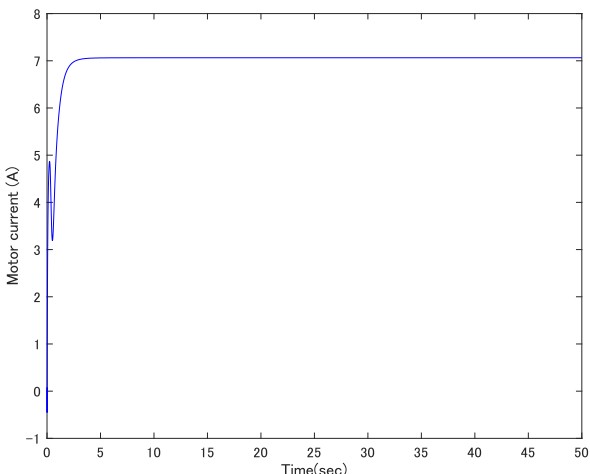

**Figure 8.** Trajectory of motor armature current $\mathcal{I}$.

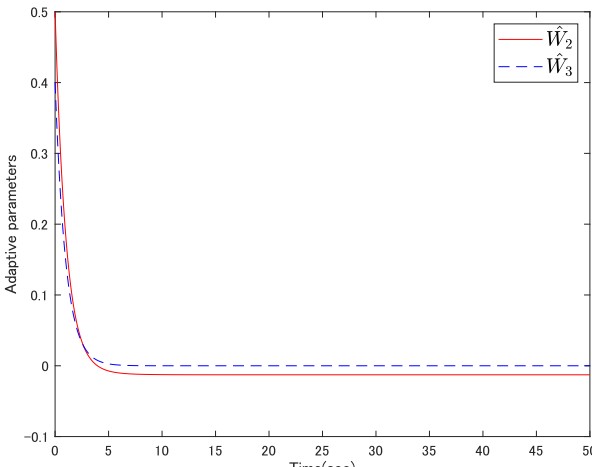

**Figure 9.** Trajectories of adaptive law $\hat{W}_2$ and $\hat{W}_3$.

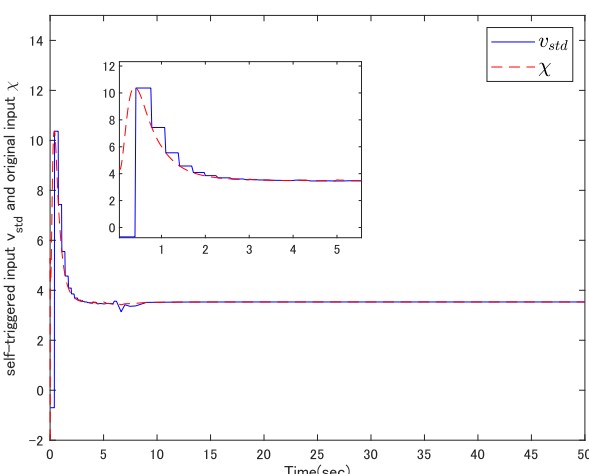

**Figure 10.** Trajectoryies of input control voltage $\mathcal{U}$ and self-triggered input $v_{std}$.

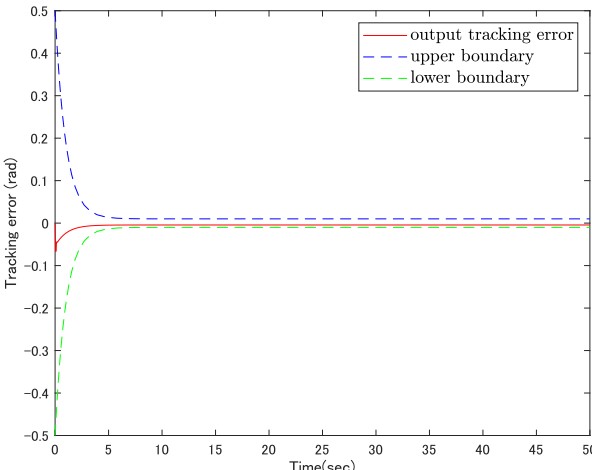

**Figure 11.** Tracking error via the command-filtered control method.

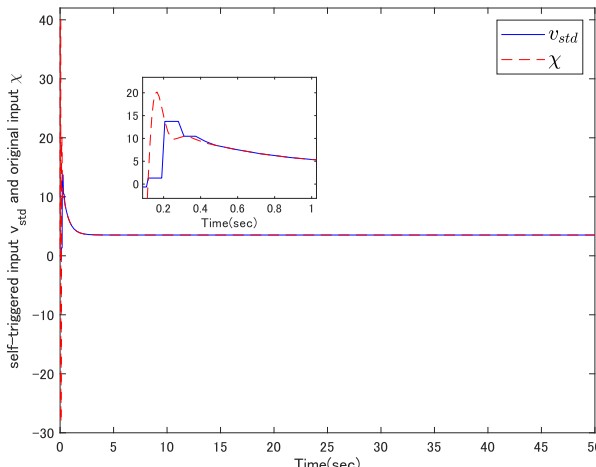

**Figure 12.** Trajectories of control input $\mathcal{U}$ via the command-filtered control method.

*5.3. Discussion*

From the simulation results, it is clear that despite the lack of information on the reference signal derivative, the boundedness of the closed-loop signal and the specified

tracking quality are guaranteed. It can be seen that the low-computation strategy in this paper requires a control input voltage of 10 Vdc. However, under the same performance specifications and initial conditions, the input voltage required by the command filtering method requires a larger control action. Furthermore, the control system can effectively achieve the tracking of the reference signal with the self-triggered input signal.

## 6. Conclusions

This paper investigates the problem of tracking control of uncertain network systems against input saturation. To solve this problem, we introduce a low-computation adaptive self-triggered control method using prescribed performance. The computational complexity was reduced by using two novel error transformation functions instead of the command filtering method. It is easier to implement in practical applications because the higher-order derivative information of the reference signal is not required. In addition, our auxiliary design system has effectively solved the input saturation problem, while ensuring that all closed-loop system signals remain bounded. This paper considers a class of strict-feedback systems, where a wide variety of engineering plants can be modeled in the form of strict feedback or can be converted to strict feedback, such as jet engine compressors, aircraft wing rocks, and single-link flexible robots. Finally, a numerical simulation and a practical simulation confirmed the effectiveness of our proposed method. Since the control strategy proposed in this paper is based on state feedback, which means that the system state is required to be completely known, our future work will attempt to extend the results to control schemes based on output feedback.

**Author Contributions:** Conceptualization, X.Z.; methodology, N.X.; software, W.W.; validation, A.M.A., N.X. and W.W.; formal analysis, B.N.; investigation, W.W.; resources, X.Z.; writing—original draft preparation, W.W.; writing—review and editing, N.X. and W.W.; supervision, B.N.; funding acquisition, X.Z. All authors have read and agreed to the published version of the manuscript.

**Funding:** This research work was funded by Institutional Fund Projects under grant no. (IFPIP: 132-611-1443). The authors gratefully acknowledge technical and financial support provided by the Ministry of Education and King Abdulaziz University, DSR, Jeddah, Saudi Arabia.

**Acknowledgments:** This research work was funded by Institutional Fund Projects under grant no. (IFPIP: 132-611-1443). The authors gratefully acknowledge technical and financial support provided by the Ministry of Education and King Abdulaziz University, DSR, Jeddah, Saudi Arabia.

**Data Availability Statement:** Not applicable.

**Conflicts of Interest:** The authors declare no conflict of interest.

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
