# Peer review of "Low-Computation Adaptive Saturated Self-Triggered Tracking Control of Uncertain Networked Systems"

_electronics, doi:10.3390/electronics12132771_

Round 1

Reviewer 1 Report

Authors are studied and presented well.

I have some questions; 1. How are the functions Φ1 and Φ2 chosen in the numerical examples, is there any condition to choose?. 2. Is it possible to extend this problem to three variables? like x1, x2 and x3. If so, try to present those results.  3. What is the computation time for proposed controls and pre existing controls.   Further, instead of choosing some particular function (Numerical examples), if authors studied real life applications it will be good for readers. 

Reviewer 2 Report

In the paper the problem of tracking control of uncertain network systems
against input saturation i considered. To solve the problem a low-computation adaptive self-triggering control method using prescribed performance is introduced. The suggested controller successfully circumvents the problem of complexity explosion found in the conventional
back stepping method and also saves communication resources without using any filters. The auxiliary design system effectively solves the input saturation problem as all of the closed-loop system signals remain bounded. The proposed method is confirmed with numerical simulations.

Reviewer 3 Report

The current paper investigates the problem of tracking control of uncertain network systems against input saturation, the problem is solved by introducing a low-computation adaptive self-triggering control method using prescribed performance. The proposed controller successfully circumvents the problem of complexity explosion found in the conventional backstepping method and saves communication resources without using any filters. The theory is validated using simulations.

Comments to authors:

- Please add more details of how the theory from the first sections is applied in the results section.

- The authors can add the steps of implementing the algorithm. The theoretical part can be better detailed. The steps will be to the benefit of the readers, maybe they’ll help the readers to implement the proposed algorithm.

- Define and detail all the parameters used.

- Specify the sampling period.

- Add both the advantages and the disadvantages of the proposed method, in the current version of the paper only the advantages are presented.

- The authors discuss about optimal design. Please define the objective function.

- Add the measurement units labels for abscissa and ordinate for all the figures from the paper.

- The state of the art it is poor regarding representative papers, maybe the author could add the following publications:

o Hybrid Data-Driven Fuzzy Active Disturbance Rejection Control for Tower Crane Systems, European Journal of Control, vol. 58, pp. 373-387-11, 2021.

o Enhanced P-type Control: Indirect Adaptive Learning from Set-point Updates, IEEE Transactions on Automatic Control, DOI: 10.1109/TAC.2022.3154347, 2022.

- Please comment on the obtained results.

Round 2

Reviewer 3 Report

The authors answered to all my concerns. From my point of view the paper can be accepted to be published in Electronics Jounral.